# Online Relinquishments of Dogs and Cats in Australia

**DOI:** 10.3390/ani8020025

**Published:** 2018-02-07

**Authors:** Susan J. Hazel, Caitlin J. Jenvey, Jonathan Tuke

**Affiliations:** 1School of Animal & Veterinary Sciences and Animal Welfare Science Centre, University of Adelaide, Adelaide, SA 5005, Australia; caitlin.jenvey@outlook.com; 2School of Mathematical Sciences, University of Adelaide, Adelaide, SA 5005, Australia; simon.tuke@adelaide.edu.au

**Keywords:** relinquishment, dog, cat, online, Australia

## Abstract

**Simple Summary:**

The aim of this study was to analyze dog and cat advertisements on a popular online trading website in Australia in February 2016. A total of 2640 ads for dogs and 2093 ads for cats were classified as being relinquished on Gumtree. A total of 23% of dog ads and 62% of cat ads were for free animals. The median age was 1.42 years in dogs and 0.9 years in cats. Compared to the human population there were proportionately more ads in Queensland and fewer ads in Victoria. In comparison to pets from animal shelters advertised on PetRescue, there were more purebred dogs on Gumtree, although the common breeds were similar. Fifteen people who had relinquished a dog or cat on Gumtree were interviewed. They used Gumtree because they believed shelters were full, they wanted to see/interview the new owner, or because they originally got the animal on Gumtree and it works. These results shed light on a hitherto under-studied population of relinquished dogs and cats.

**Abstract:**

While traditionally people relinquish their pets to an animal shelter or pound, the internet provides a newer method to re-home. We analyzed advertisements (ads) on the largest website in Australia for trading dogs and cats: Gumtree. Data was collected in 2016. Dogs were sampled on 7, 16 and 24 February 2016 and cats on 9, 19 and 26 February 2016, with 2640 ads for relinquished dogs, and 2093 ads for relinquished cats. It was estimated >31,000 puppies/dogs and >24,000 kittens/cats are relinquished on Gumtree per year. The median age of dogs was 1.42 and cats 0.9 years of age. There were 23% of dog ads and 62% of cat ads for free animals. Compared to the human population, there were proportionately more ads in Queensland and fewer ads in Victoria. A total of 15 people were surveyed who had relinquished a dog or cat using Gumtree. The dog owners used Gumtree for two reasons: because they believed the shelters were full (n = 4); and they wanted to see/interview the new owner (n = 2). For cat owners: they had originally got the cat on Gumtree (n = 2); they use Gumtree for other things, and it works (n = 2), and; they wanted to see/interview the new owner (n = 2). The data collected will be valuable for implementation of policy and interventions to protect the welfare of unwanted dogs and cats.

## 1. Introduction

Australia has an owned dog and cat population of approximately 4.8 million and 3.9 million, respectively [1]. However, it is likely that the total population of dogs and cats is higher, as un-owned animals/strays, and those in registered or independent shelters or pounds would also contribute to the total population. In addition, the ownership of an animal may be transferred through sale or other trade, and animals in a shelter or pound can be rehomed. This results in a complex network of animals transferred between the owned and un-owned populations, and between different owners. Animals leave the care of their owners through being lost, surrenders to a shelter or pound, being abandoned, being given away or being sold. Worldwide, there have been attempts to measure the un-owned population of dogs and cats in shelters [2,3], but there is a lack of research into the dogs and cats that are transferred between owners and between owned and un-owned populations in other ways.

Much of what we know about the flow of animals from the owned and un-owned populations comes from rescue shelters and pounds. Larger rescue organizations collect data on numbers of surrendered dogs and cats each year. There were 55,570 cats admitted to Royal Society for the Prevention of Cruelty to Animals (RSPCA) shelters around Australia in 2015/16, with numbers rising in the last few years. In contrast, there were 45,256 dogs admitted to RSPCA shelters in the same year and numbers are declining [4]. However, the RSPCA represents only a portion of the surrendered population. The total number of both stray and surrendered dogs in Australia in 2012/13 from shelters and local municipal facilities, has been estimated at 211,655 dogs admitted, with 101,037 of these reclaimed [2]. There is no data available estimating the total numbers of stray and surrendered cats in Australia, although it is likely to be even higher. The incidence of unowned dogs and cats is a worldwide problem, with an estimated 129,743 dogs and 131,070 cats entering UK shelters in 2009 [3], and a study from South Korea reporting over 10,000 abandoned dogs in Seoul [5]. In the US, it was estimated that 4.4% of dog-owning and 3.8% of cat-owning households had relinquished a pet to a shelter in the previous year [6].

The relinquishment of pets represents a significant cost to society in a number of ways. In Australia in 2004, there was an estimated $AUS180 million spent annually by animal welfare agencies [7], while in the United Kingdom expenditure by animal welfare organizations was approximately £340 million in 2010 [8]. Local governments in Australia in 2004 spent $AUS83 million on animal management [7]; in the UK in 2011 the comparable figure was £57.5 million [9]. While local government costs include dog registrations and dealing with dog attacks, a large portion of these figures would be expected to arise from stray and relinquished animals.

As well as the economic costs there are also emotional costs as people may be forced to relinquish their pets due to external influences, such as a relationship breakdown or inability to find rental accommodation that will allow a pet [10,11]. There are also emotional costs experienced by staff involved in euthanizing large numbers of pets at shelters, with shelter workers struggling with the moral dilemma of their job, often resulting in mental health problems and high staff turnover [12,13,14]. Finally, un-owned dogs and cats are at an increased risk of impaired welfare using both physical and psychological measures of welfare [15,16], as well as the emergence of diseases, demonstrated by an outbreak of virulent systemic feline calicivirus in cats [17], and of *Streptococcus equi* pneumonia in dogs [18].

If we are to design evidence-based strategies to reduce the numbers of dogs and cats relinquished, as well as protect the welfare of pets in which relinquishment cannot be avoided, we need to know more about the population of dogs and cats involved. Some shelters survey owners who are relinquishing pets to determine their reasons for relinquishment [19,20], and also collect demographic data on the age, sex and breed of the relinquished animal [4,19,20]. While this information is not perfect (for example, owners may not be truthful about the behavior of their pet if they think it will negatively impact the likelihood of that pet finding a new home), it has been helpful in designing interventions. For instance, knowing that sometimes pets are relinquished because owners cannot find rental accommodation that allows pets has resulted in real changes, with the Australian state of Victoria recently announcing modifications to rental agreements to make it easier for people to rent with pets [21].

Pets are given away or sold through advertisements in newspapers, bulletin boards and using the internet. In Australia, Gumtree [22] has the highest volume of online ads for dogs and cats, followed by Trading Post [23]. Breeders of purebred dogs also advertise on DogzOnline [24], although this site represents a smaller number of animals compared to Gumtree and Trading Post. Other forms of social media, such as Facebook^®^, also advertise pets for sale or to find new homes. These online sites advertise both puppies and kittens being sold by a breeder, and animals being relinquished.

There has been limited research into online trading of dogs and cats. In a US study, dog breeders who advertised puppies on the internet were less knowledgeable about health issues specific to their breed/s and less likely to screen their animals for heritable diseases compared to dog breeders who did not advertise their puppies on a puppy internet site [25]. To our knowledge, no published studies have focused on pets being relinquished online. The aims of the present study were to: (1) estimate the total numbers and prices of dogs and cats relinquished online on Gumtree; (2) analyze the breed and Australian State/Territory of origin of dogs and cats relinquished on Gumtree and compare with animals presented to PetRescue [26] (an online charity advertising pets from a shelter or rescue organization in Australia) and the RSPCA; and (3) interview a sample of pet owners relinquishing their pets on Gumtree to determine their reasons for relinquishment and why they chose to advertise their pet online.

## 2. Materials and Methods 

### 2.1. Extraction of Data

Data from Gumtree were collected over approximately three weeks for both dogs and cats: 8, 17 and 25 February 2016 for dogs and 9, 19 and 26 February 2016 for cats. Data were extracted using the BeautifulSoup module in Python Version 3.2.1 (Python Software Foundation, Wilmington, DE, USA) to parse the ads, then cleaned and analyzed using R (The R Foundation, Vienna, Austria) and Pivot tables in Microsoft Excel^®^ 2013 (Microsoft, Redmond, WA, USA). Cleaning was completed within Microsoft Excel^®^ 2013 by scrutinizing individual ads to remove duplicates or ads that were not for dogs or cats (e.g., equipment, pet sitting services). Data for the dog and cat breed and cross/purebred status, owner status (owner/breeder), age, price and State/Territory were collected. Where the town, but not the State or Postcode, were provided, an internet-based Postcode finder was used to determine the State/Territory. If there was more than one place with the same name in more than one State/Territory, then the ad was left as State/Territory unknown. An estimate of the number of dogs and cats in each ad was made by generating random numbers using R and then analyzing the average number of dogs/cats per ad in those random ads. There were 50 random ads used for both dogs (1.9% of total ads) and cats (2.4% of total ads). Data on neuter, microchip and vaccination status were not collected as a minority of ads reported on these factors.

### 2.2. Analysis of Relinquishment Ads

Dogs and cats advertised on the internet represent a mixture of puppies/kittens being sold alongside pets being relinquished or traded. Many ads request a sum of money for the pet, even when it appears the pet is being relinquished. The definition of relinquishment used for this study was the Oxford definition: ‘Voluntarily cease to keep or claim; give up’ [27]. Dogs were defined as being relinquished if: (1) there was a statement within the ad confirming that a new home was being sought (e.g., seeking new home, very sad to part with, need to find a new forever home); (2) they were being offered for free at any age; or (3) if the dog was older than 16 weeks of age and there was no indication it had been advertised as a puppy and was still looking for a home (‘‘X’ is a 6 year old purebred Sheffield Blue Heeler. He is very friendly and good with other dogs and children. Microchipped and wormed. To good home only.”). On Gumtree sellers are classified as owner, breeder or shelter and the shelter animals were not included in the present study. It is possible that some very small shelters were advertising as an owner rather than as a shelter, but there was no way to differentiate these types of sellers. Both owner and breeder categories were included as there were some breeders who were relinquishing pets using the above criteria (e.g., “Owner breeder out of work and no permanent (sic) home, current home being sold by owner, cannot find another suitable home.” or “Three female blue amstaffs—a mother and her two daughters—I’ve raised. My intentions were to bread (sic) them but things have changed. They’re pedigree papered registered ect (sic). Spent alot $ for these lines call for more info ect (sic).”). The same criteria were used to determine a cat being relinquished. In some ads there was more than one animal advertised. In those cases, data was collected only from the first animal mentioned in the ad (e.g., breed, age, price), and the other animal/s were not included.

The relinquished ads on Gumtree per State/Territory were compared with data provided by John Bishop, Co-Founder and Joint CEO of PetRescue, for the same time period as the Gumtree data were collected, published figures on total dog shelter admissions to shelters and councils in 2012/13 [2] and published RSPCA statistics for cats from 2015/16 [4].

### 2.3. Surveys of Owners Who Had Relinquished Dogs or Cats

An email was sent through Gumtree informing people about the study that we were conducting and asking if they would be willing to participate. Ads were chosen from those available online using the criteria for identifying a relinquished dog, outlined above. These were selected starting from the first ads shown on each day. Due to the high numbers, not all ads for relinquished dogs or cats were included on each day. A total of 324 emails were sent to dog owners between 11 May and 14 June 2016 and 299 emails to cat owners between 17 May and 14 June 2016. The owners responded to the email either by return email or by phone. Surveys were performed either on the phone or in written form via email, depending on the preference of the respondent. Questions included: the demographics of the animal (age, breed, sex, neuter status, microchip); two Likert type scale questions on satisfaction with the health and behavior of the animal (1 is very dissatisfied and 5 is very satisfied); Where did you get pet’s name from originally?; What was the main reason you advertised pet’s name?; Why did you decide to advertise on Gumtree instead of going to a shelter; Did you advertise pet’s name anywhere else?; Have you successfully found a new owner for pet’s name yet?; If yes, do you know anything about the new owner? Participants were also given an open-ended question at the end of the survey to add any other details they wished to share. Approval for this study was provided by the University of Adelaide Human Ethics Committee (H-2016-036).

### 2.4. Statistical Tests

Comparisons between groups where the data was not normally distributed were made using Mann-Whitney tests. Categorical data was compared used Chi-square tests. Data are presented as Mean ± SEM unless otherwise stated. A significance level of p < 0.05 was used for statistical testing.

## 3. Results

### 3.1. Dogs on Gumtree

A total of 2640 individual ads were identified for dogs being relinquished on Gumtree between 7 and 24 February 2016. There were 1328 ads on 7 February, 1411 ads on 16 February and 1450 ads on 24 February. Of these, 650 were new ads on 16 February and 663 were new ads on 24 February, giving an estimate of 82 new ads/day, or by extrapolation (×365) 29,930 ads per year. The average number of dogs per ad in 50 random ads was 1.06 (all ads were for single dogs excepting 3 ads for two dogs), giving an estimate of 31,726 dogs advertised on Gumtree per year.

Most of the dogs on Gumtree were classified as purebred (n = 1361, 51.4%) with 1188 (44.6%) cross-bred dogs, and 103 (3.9%) ads in which it was not clear if dogs were pure or cross-bred. The most common dog breeds advertised on Gumtree and on PetRescue in the same time period are shown in Table 1. The top two breeds in order for both Gumtree and PetRescue were the Staffordshire Terrier and Kelpie, representing just over 20% of the total ads for both sources. The third most common breed for Gumtree was the American Staffordshire Terrier (#4 for PetRescue), with the Greyhound the third most common breed for PetRescue (#24 for Gumtree). Unlike Gumtree, approximately three quarters (75.8%) of the ads on PetRescue were for cross-breed dogs, with only 23.8% representing purebred. 

Most dogs on Gumtree were offered by the owner (n = 2480, 93.5%) with 172 ads from breeders (6.5%). The median age for the dogs was 1.42 years, with a range of 0.2–14.2 years (n = 2640; Table 2). The most common age group was 1 to 2 years (32% of ads), with dogs that were 6 months to one year of age and two to five years of age both representing 26% of the ads. Dog ages were not normally distributed, with a right skew; with 10% of the ads for dogs over 5 years of age and 21 of the ads for dogs of 10 years of age or more. Dogs that were cross-bred were advertised at a significantly lower age than purebred dogs (2.0 ± 1.9 years, median 1.3 years) versus 2.3 ± 2.0, median 1.6 years, respectively (p < 0.0001, Mann-Whitney).

The median price for each dog was $AUS200, with a range from 0 to $AUS7000. Almost one quarter (n = 575, 23%) of dogs were advertised for free, with 43% of dogs advertised for $AUS100–499 (Table 2). There were 147 ads in which no price was given. Purebred dogs were advertised for a significantly higher price than cross-bred dogs ($AUS562 ± 727 vs. $AUS198 ± 266, respectively; p < 0.0001, Mann-Whitney). 

The majority of Gumtree ads were from NSW (35.6%) and Queensland (34.8%), giving a total of approximately 70% of all Gumtree ads in Australia coming from those two States (Figure 1). The proportion of ads from Gumtree and PetRescue and published figures on total dog shelter admissions in 2012/13 [2] were compared with the State/Territory human population sizes, using data from the Australian Bureau of Statistics [28] There was a significant difference between the State/Territory human populations and the proportions of dogs advertised on Gumtree (χ²(7) = 566.6, p < 0.0001), PetRescue (χ²(7) = 507.4, p < 0.0001) and the total dog shelter admissions [2] (χ²(7) = 4345.2, p < 0.0001). Compared to the proportion of the State/Territory human population, there were more ads for Gumtree in Queensland and NSW and fewer ads for Gumtree in Victoria. Queensland had a higher proportion of all three (Gumtree, PetRescue and total dog shelter admissions) versus the State human population (Figure 1). The proportions of dogs advertised were significantly different between Gumtree and PetRescue (χ²(7) = 158.9, p < 0.0001).

The per capita values for each marketplace and State/Territory are shown in Figure 2. The per capita number of Gumtree ads are highest for the Northern Territory and lowest in Victoria, while PetRescue and total surrender figures per capita are highest in Queensland. Per capita rates for total surrenders are higher than for Gumtree and PetRescue. The ACT per capita figures for Gumtree, PetRescue and total surrenders are 0.04, 0.04 and 8.7 per 1000 residents, respectively.

The prices of advertised dogs in each State/Territory are shown in Table 3. There was a significant difference in price category between states χ²(12) = 89.7, p < 0.0001 (data for the Australian Capital Territory (ACT) was not included as there were insufficient cases per category for analysis). The highest proportion of free dog ads was in Queensland (34.5%) followed by the Northern Territory (31.9%) with the lowest proportion of free ads in Western Australia (22.3%). The highest proportion of ads for dogs costing >$AUS500 was in Victoria (47.6%) with the lowest proportion in Tasmania (16.3%).

### 3.2. Cats on Gumtree

A total of 2093 individual ads identified as cats being relinquished were posted between 9 and 26 February 2016. This included 1353 ads on 9 February, 1397 ads on 19 February, and 1465 ads on 26 February. There were 408 new ads on 19 February and 332 new ads on 26 February, giving an estimate of 44 new ads per day or 16,060 ads per year (44 × 365). The average number of cats per ad in 50 random ads was 1.51 (37 ads for single cats/kittens, 8 ads for two cats/kittens, 3 ads for five cats/kittens and one ad for six kittens; no number provided in one ad), giving an estimate of 24,250 cats advertised on Gumtree per year.

A minority of 508 ads provided any information on whether a cat was a pure or cross-breed, with 61% of these purebred (n = 310) and 39% cross-breed (n = 198). The breeds for which there were at least 15 ads on Gumtree are shown in Table 4 in comparison to cats advertised on PetRescue in the same time period. The Ragdoll and Domestic Shorthair were the most common breeds mentioned in the Gumtree ads, however, Domestic Shorthair and Domestic Medium Hair were by far the most common breeds on PetRescue.

Most cats were offered by the owner (n = 2003, 95.7%) with 90 ads from breeders (4.3%). The median age for the cats was 0.9 years, with a range of 0 to 18.2 years (n = 2093). The most common age group was up to 16 weeks (23.6% of ads) with around half of the ads (53%) for cats that were one year of age or less (Table 5). Cat ages were not normally distributed, with a right skew; 10% of the ads were for cats over 5 years of age, and there were 9 ads for cats of over 15 years of age. Cats that were cross-bred were advertised at a significantly lower age than purebred cats (2.2 ± 2.7 years, median 1.3 years versus 2.5 ± 2.5, median 1.6 years, respectively; p < 0.01, Mann-Whitney).

The median price for each cat was $AUS0, with a range from 0 to $AUS2500. Almost two thirds (n = 1298, 62%) of cats were advertised for free, with 30% of cats advertised for $AUS1–499 (Table 6). Purebred cats were advertised for a significantly higher price than the cross-bred cats ($AUS351 ± 331 vs. $AUS66 ± 99, respectively, n = 508; p < 0.0001, Mann-Whitney).

The same proportion of cat ads on Gumtree were from NSW and Queensland (32.7% each; Figure 3). The proportion of cat ads from Gumtree and PetRescue were compared with the State/Territory human population sizes [28]. There was a significant difference between the State/Territory human populations and the proportions of cats advertised on Gumtree (χ²(7) = 566.6, p < 0.0001), PetRescue (χ²(7) = 507.4, p < 0.0001) and the RSPCA (χ²(7) = 9969.9, p < 0.0001). In Queensland, the proportion of cats from all three sources was higher than the Queensland human population, while in Victoria PetRescue and Gumtree represented a lower proportion than the State human population. In WA Gumtree, PetRescue and the RSPCA were a lower proportion than the human population, while in NSW only the proportion of PetRescue ads was substantially lower than the State human population. The proportions of cats advertised was significantly different between Gumtree and Petrescue (χ²(7) = 158.9, p < 0.0001). Compared to PetRescue, there were more Gumtree ads in NSW, SA, Tasmania and the Northern Territory than expected, and fewer ads in Victoria than expected.

The per capita values for each marketplace and State/Territory are shown in Figure 4. The per capita number of Gumtree ads are highest for the Northern Territory and lowest in Victoria, while PetRescue and RSPCA figures per capita are highest in Queensland. Per capita rates for the RSPCA are higher than for Gumtree and PetRescue. The ACT per capita figures for Gumtree, PetRescue and the RSPCA are 0.08, 0.12 and 6.3 per 1000 residents, respectively. The per capita rate of cat admissions to RSPCA shelters is higher in the ACT than any other State/Territory.

The prices of advertised cats on Gumtree in each State/Territory were compared using three categories: $AUS0, $AUS1–499 and >$AUS500. The data for the ACT, NT, and Tasmania were not included as there were insufficient cases per category. There was a significant difference in price category between states χ²(8) = 38.6, p < 0.0001. The highest proportion of free cat ads came from Victoria (77%) and SA (67%) with the lowest proportion of free ads in WA (54%; Table 6). Victoria had a lower proportion of Gumtree ads for cats $AUS1–499 (16% vs. >30% for other States), and the highest proportion of ads for cats costing >$AUS500 (7%). The proportions of Gumtree ads for cats >$AUS500 was low overall (3–7%).

### 3.3. Interviews

Most interviews took place on the phone (n = 11), with four people emailing back a survey. There were 15 responses in total, a response rate of 2.5% (8/324) for dogs and 2.3% (7/299) for cats. In addition, 16 people responded to the email to notify that they would not be involved, giving reasons such as not having time, or that the situation was too distressing.

There was a mix of dog and cat breeds being relinquished by the respondents (Table 7). Five of the dog owners had desexed dogs, with two dogs entire, and one in which the response was unclear. All dogs were microchipped, excepting one in which the response to this question was also unclear. Six dog owners scored a 4/5 (satisfied or highly satisfied) with their dog’s health. Five dog owners also responded on their dog’s behavior, with three of the five highly satisfied and two giving a 3 or 3.5, i.e., neither satisfied nor dissatisfied. Four of the dogs were originally obtained online – three from Gumtree and one from TradingPost. Only one cat was entire, with the owner not clearly stating neuter status in one other cat. All cat owners were highly satisfied with their cat’s health, but one owner scored their cat’s behavior a 2/5 and another owner 3/5 (Table 7). 

There were a variety of reasons given for relinquishment of the dogs and cats (Table 8). For the eight dogs, the most common reasons were that they had rescued the dog and needed to find a new home for it (n = 2), and the behaviour of the dog (escaping in one dog and too interested in nearby livestock in another). For two dog owners, two reasons were provided for the relinquishment, in the other six cases, a single reason was provided. For the seven cats, the most common reasons for relinquishment were that the pets in the house were not getting on (cats in two cases and a dog in one case) and moving (n = 2). For three of the cat owners, two reasons were provided for the relinquishment, in the other four cases, a single reason was provided.

When asked the reason for using Gumtree rather than other places, the most common reason for the dog owners was that they believed the shelters were full (n = 4) and that they wanted to see/interview the new owner of their dog (n = 2; Table 9). The cat owners chose Gumtree as they originally got the cat on Gumtree (n = 2), because they use Gumtree for other things and it works (n = 2) and to see/interview the new owners of their cat (n = 2). There were three dog owners and four cat owners who gave two different reasons to choose Gumtree.

Relinquishing owners were also asked about the success of rehoming their pet on Gumtree. Of the eight dog owners, three had successfully rehomed their dog at the time of completing the survey, and another owner had a prospective re-home organized. In one of the successful re-homings, the owner had carried out two failed trials with the dog, with a third trial being successful. For one dog owner there had been two responses with neither suitable, and in another, there had been so many responses that they had had to take the ad down, although none had been successful at the time of the interview. In another two dog owners there had not yet been any responses. For the seven cat owners, one person had had 50 offers for the cat within ten minutes, and another had two potential homes within one day. Both of the cats in these examples were purebreds. In another two cat owners there had been a single response, the three remaining owners had not yet received responses.

## 4. Discussion

The current study provides evidence about the numbers and types of dogs and cats relinquished on one popular website Gumtree in Australia, filling an important gap in knowledge. It indicates that thousands of dogs and cats are being relinquished on Gumtree per year. Information collected on their breed and age profiles allows comparison with other relinquished populations on PetRescue and published RSPCA and surrender data [2,4]. Although there are similarities between the populations, such as the most common dog breeds on Gumtree, PetRescue and in published figures, there also appear to be differences suggesting pets relinquished on Gumtree represent a subpopulation of relinquished animals.

By extrapolation, the total number of dogs and cats relinquished via Gumtree ads was approximately 30,000 and 16,000, respectively. Recent estimates have indicated that 211,655 dogs were admitted to shelters and municipal facilities in Australia in 2012/13 [2]. Some of the dogs being offered on Gumtree may eventually be relinquished to a shelter or end up at a municipal pound if a new owner is not found. The estimated Gumtree ads represent a significant proportion (~14%) of the total number of dogs admitted to shelters and municipal facilities per year. This proportion would be even higher if one considered other online websites (e.g., DogzOnline, TradingPost), as well as Facebook^®^, and other private or closed websites advertising pets. The diverse and diffuse nature of online ads, as well as the fact some are closed websites, makes it impossible to estimate the total numbers of pets advertised online. If it is also considered that less than half of shelter relinquishments are owner surrenders (19% in dogs to Queensland RSPCA shelters in 2014 and 32% of cats between 2006 and 2009 in Queensland RSPCA shelters; [19,29]) then the online ads would represent an even higher proportion of owner surrenders.

There is a complex flow of dogs and cats between the owned and un-owned populations. The data presented from the Gumtree ads, from Pet Rescue (representing larger and smaller shelters and pounds) and from published data from the RSPCA, illustrate the complexity of movements of surrendered dogs and cats within and between Australian States/Territories. The human population for each State/Territory was used for comparison; if one assumes that a similar percentage of people in each State/Territory own a dog or cat, the human population is a proxy for the dog and cat population. If one then assumes that a similar proportion of dog and cat owners in each State/Territory relinquish their dogs or cats each year, the proportions should be similar. In fact, they were not, and this may be because these two assumptions are not true, and that people in different States/Territories are more or less likely to own a dog/cat and to then relinquish it. In general, Queensland had an overrepresentation of the surrendered population of dogs and cats on Gumtree, PetRescue and the RSPCA compared to their human population, while Victoria tended to have an underrepresentation of relinquished dogs and cats compared to their population. An additional confounder is that RSPCA data is dependent on the numbers and sizes of shelters in each State/Territory, and the PetRescue data depends on the market penetration of shelters/pounds using their website from each region. Internet access and hence access to Gumtree may be more uniform between States. To fully understand the flow of owned and un-owned dogs and cats within and between regions of Australia will require further research.

Although there were similar total numbers of ads for dogs and cats on Gumtree, there were fewer new ads posted in the follow-up datasets for cats versus dogs. This suggests the turnover of cat ads on Gumtree is longer than for dogs. Of the dog and cat owners that were interviewed, a higher proportion of dog owners had already successfully rehomed their pets versus the cat owners, at the time of interview. Of the two cat owners who had multiple responses for relinquishment, and had been able to successfully rehome their pets, both were purebred cats. However, the very small number of responses from owners relinquishing their pets on Gumtree means that this data is not likely to be representative of the entire population of relinquishing owners.

There are many risks involved with online trade in pets for both prospective owners and animals, but as yet online sales of dogs and cats remain unregulated around the world. In some countries voluntary standards for online sales of pets have been developed, with the Pet Advertising Advisory Group in the UK, the Irish Pet Advertising Advisory Group and the BelgPAAG (Belgian Pet Advertising Advisory Group) was recently launched in Belgium [30]. The EU Dog & Cat Alliance released a report into the cost of online sales in the EU recently [31]. The main findings were that online ads are now the most common way for people to purchase a pet, that there are no regulations around Europe that cover this trade, and the many risks involved. These include sale of unweaned animals and animals in poor health. There are up to 269,620 dogs and 67,847 cats estimated as being advertised online in the European Union on any given day [31]. A report issued by the Better Business Bureau (BBB)^®^ International Investigations Initiative presents evidence on how online pet sellers scam pet buyers in the US [32]. The BBB ScamTracker contained 907 reports on pet scams at the time the report was published, which was 12.5% of all online purchase fraud complaints. Many of the scams do not involve a real animal, for example, an ad for a free dog or cat that urgently needs a new home may be posted. Once a person responds to the advertisement, costs relating to transportation and care of the animal are requested. The report includes information from other countries, with 337 pet complaints to the Australian Competition and Consumer Commission in the first six months of 2017, and 377 complaints involving animals to the Canadian Antifraud Centre in 2016, with estimated losses of $222,000 [32].

In addition to the risks relating to online sales of dogs and cats outlined above, there are other risks relating specifically to dog and cat relinquishment. If an aggressive dog is relinquished to a reputable shelter, the dog’s behavior will be assessed and the risk to new owners evaluated. Although behavioral assessments performed in shelters are unreliable [33], prospective owners can be educated on the behavior of the dog and triggers of aggression prior to adoption, and post-adoption support provided if behavioral problems arise. In addition, good shelters have access to trainers and behaviorists who can implement programs to improve the behavior of an animal. In some of the ads it was possible to read between the lines and see that aggression was likely to have been a problem (e.g., “Urgent rehoming needed for Jack Russell cross, male, 10 years. Beautiful dog, very friendly but best suited to a kid free home.”) but there is nothing to prevent owners advertising and selling pets with a behavioral problem without providing any warning or information to the new owner.

As well as the risk to new owners, there are also risks to the animals themselves. While not perfect, shelters do question prospective new owners on their ability to look after a pet and may not allow somebody to adopt a pet if they believe it will not be well cared for. Again, there is no obligation when dogs and cats are traded online for this to occur. As people may have ready access to the internet at any time of the day, and with the posting of cute photos, impulse buying may also be a significant problem. The wording of some of the ads indicated an impulse purchase, for example: “6-month-old Shar Pei puppy we brought him off Gumtree two weeks ago and he's the best puppy ever but unfortunately I have no time for him as I have three babies. He has had all he's (sic) needle are up to date. Looking for a good home with someone who has plenty of time for him”. An additional risk associated with the Gumtree ads was that a large proportion were free, or a minimal cost. It has been suggested that animals obtained at no or a low cost are at an increased risk of future relinquishment [34], perpetuating the problem. However, a later study suggested that the attachment of people to cats did not differ between free- and fee-based cat adoptions [35]. Further research will be needed to assess whether free animals adopted online are more or less at risk of being re-relinquished.

There is also an additional risk associated with offering dogs and cats free, or very low cost. One of the interview respondents discussed how they had been taking home dogs advertised on Gumtree free in order to find them a new home. This respondent felt that otherwise, people involved in dog fighting would take them (they had rehomed approximately 80 animals in the past year). To our knowledge, there is no evidence of trade and use of dogs in dog fighting, although it should be considered as a possibility. There were a number of ads describing dogs suitable for pig hunting, which is illegal in some States of Australia [36]. In fact, the Bull Arab, which is not a recognized breed by the Australian National Kennel Council [37], is a breed often associated with pig hunting, and a large number of Gumtree ads for Bull Arabs from Queensland (in which pig hunting is legal) were observed. The final risk for dogs and cats advertised free or for low cost online is from animal hoarders. Arluke et al. (2017) describe three types of animal hoarders: the overwhelmed caregiver, rescuer, and exploiter. The rescue hoarder has a missionary zeal to save all animals and actively seeks to acquire animals [38]. Being exposed to advertisements of animals that need rescuing on Gumtree is likely to trigger a rescue hoarder, who can acquire multiple animals free or at minimal cost.

It is difficult to determine if the population of dogs and cats surrendered on Gumtree overlaps with the animals that are surrendered to the RSPCA or other shelters. Four of the people interviewed who had relinquished dogs on Gumtree stated that they used Gumtree as the local shelters were full, and while this was not given as a reason by any cat owners there was one person who said they used Gumtree because the cat would be ‘put down’ at a shelter. The animals advertised on Gumtree would represent the ‘owner surrender’ dog admissions to a shelter. Only 19% of the 11,967 dogs entering the RSPCA in Queensland in 2014 were owner surrender, with 24% classified as strays admitted by the public, and 34% from municipal councils [29]. However, some of the strays admitted by the public may represent owned animals, with owners avoiding paying shelter surrender fees, or not wanting people to know they are relinquishing their own pet. The top four breeds processed by RSPCA shelters in Queensland in 2014 were the Staffordshire Bull Terrier (20%), Australian Cattle Dog (8%), Kelpie (7%) and Bull Arab (7%). The top four breeds advertised on Gumtree in the current study were the Staffordshire Bull Terrier (12%), Bull Arab (8%), Kelpie (6%) and Australian Cattle Dog (5%). Thus, the most popular breeds for the RSPCA and Gumtree are similar, and also to the dogs offered on PetRescue which represent both larger and smaller shelters. Despite similarities in the breeds, the proportions of pure and cross-breed dogs differ widely, with 92% of the dogs admitted to the RSPCA in Queensland in 2014 being cross-bred, while only 46% of dogs from Queensland advertised on Gumtree cross-bred. The median price of purebred dogs was significantly higher in the Gumtree ads than for cross-bred dogs.

There is an important difference in relinquishing online versus giving up a dog or cat at a shelter. At a shelter, owners may have to pay a fee to relinquish their pets, whereas on an online site, they can request a sum of money. However, based on the definition of relinquishment being used in this study (“Voluntarily cease to keep or claim; give up”) the owners are voluntarily giving up their dog or cat. Indeed, if a dog or cat has been purchased originally for a large sum of money, offering the pet online may allow the owners to make up for some of this loss. The general use of the term relinquishment describes an animal taken to a shelter. When devising the criteria to determine if a dog or cat was being relinquished, we originally planned to have a cap on price for the animals we included in our study. However, when reading the descriptions of the ads, there were ads in which the dog or cat was being traded for over $AUS1000, and yet the wording suggested that the decision of the owner to relinquish their pet had been difficult. Thus, it was not possible to use a cut-off price and say that all pets below that price were relinquished and above that price were not.

One reason provided by respondents for the use of Gumtree for relinquishing their pets was that they could speak to the new owners and see where their pet would live before agreeing to give their pet a new home. One of the owners had even run trials of their dog with the prospective new owners and had two failed trials before having success with the third. Follow-up and control on the re-homing of their pet is something that cannot be provided in a traditional shelter, where the animal is left and no further information on where, and even if it has been rehomed, is provided. A novel dog adoption program has been suggested in the US which involves placing dogs into foster homes, with the foster carer tasked with finding the dog a new home [39]. There are obvious issues with protection of privacy and the possibility of the new owners being harassed by the previous owner (or vice versa), but this feedback is important for shelters to consider. To accommodate owners who want to know where their pet goes and that it is being well looked after, innovative programs should be considered.

An important limitation of the current study was the limited number of relinquished owners who responded. Of those who did respond, all would be classified as responsible pet owners who were in positions in which they had no real choice but to relinquish their pet, and who were doing their best to find a good home. One of the respondents had rescued the animal previously from an ad on Gumtree and believed that the owner was going to kill the animal if a new owner was not found quickly (an unavoidable change in this persons’ living arrangements outside of their control meant that they needed to relinquish the pet). It is likely the population of people relinquishing their pets on Gumtree are mixed, with some doing everything they can to find a good home, and others not caring where their pet goes as long as they can dispose of it. Further research with responses from a larger range of respondents would be difficult, as people who do not care for their pet are also unlikely to respond to requests to participate in research. For the owners who were relinquishing their pets, several were obviously upset about the difficult decision they had been forced to make. It was also interesting that some owners felt it necessary to respond to the email to notify us that they were not willing to participate as they found the situation too distressing. This aligns with previous research that concluded that rather than giving their pet up at a shelter thoughtlessly many people struggled with an unavoidable situation and really had no choice [20].

Research based on the use of big data from web scraping (automated collection of data from web pages) provides both important research data but also ethical questions. Guidelines for internet research have been published [40], and we believe the current study presents no significant ethical problem using these guidelines. The respondents in the present study who were interviewed provided their informed consent to participate, and no personal details are included that could be used to identify them. However, owners whose data were used to estimate numbers and demographics of the relinquished pets did not consent to this use. Although the data were publically available on Gumtree, it can be argued that people did not consent to other uses of their information. However, risks to these people are minimal in the present study as aggregate data were used that do not permit identification of individuals, and the direct quotes used in the paper are unlikely to be able to be used to identify individuals as the ads are now more than a year old and are taken down from Gumtree once new owners are found, or owners decide not to continue advertising. The benefits of the research in understanding a population of relinquished pets that have hitherto not been studied are also likely to outweigh the minimal risks to the individuals whose data was used.

Another important limitation of the study is that not all information posted by the dog and cat owners is accurate, for example, some owners may not know the true age of their pet or may provide false information on factors such as dog breed. Owners may also provide false reasons for the relinquishment due to social pressure to have others think well of us [41]. Finally, the extrapolation from the ads posted in February to the total number per year is likely to be inaccurate as numbers of ads will vary from month to month. This is particularly relevant for cats, as they are seasonal breeders.

## 5. Conclusions

The present study provides evidence that many thousands of dogs and cats are relinquished online in Australia each year. This population of pets has previously not been considered in policy or strategies to reduce the risk of relinquishment and protect the welfare of un-owned dogs and cats. Knowledge of online relinquishments adds detail to what we currently know about the complex flow of dogs and cats between owned and un-owned populations, and between different owners.

## Figures and Tables

**Figure 1 animals-08-00025-f001:**
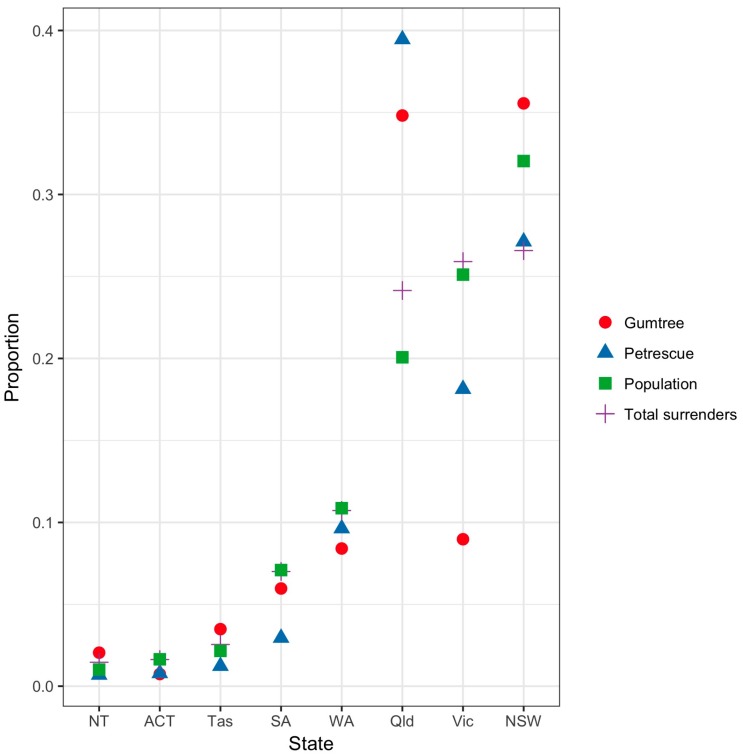
Proportions of total ads for relinquished dogs on Gumtree and PetRescue from 8 to 24 February 2016 and total dog shelter admissions from Chua, Rand and Morton [2] for each Australian State/Territory. States/Territories are ordered in increasing proportion of total population. The square symbol is the proportion of the Australian population for each State/Territory. (Note: The proportion of the Australian population in each State/Territory is: New South Wales (NSW) 31.7%, Victoria (Vic) 24.9%, Queensland (Qld) 19.9%, Western Australia (WA) 10.8%, South Australia (SA) 7%, Tasmania (Tas) 2.1%, Australian Capital Territory (ACT) 1.6% and Northern Territory (NT) 1%).

**Figure 2 animals-08-00025-f002:**
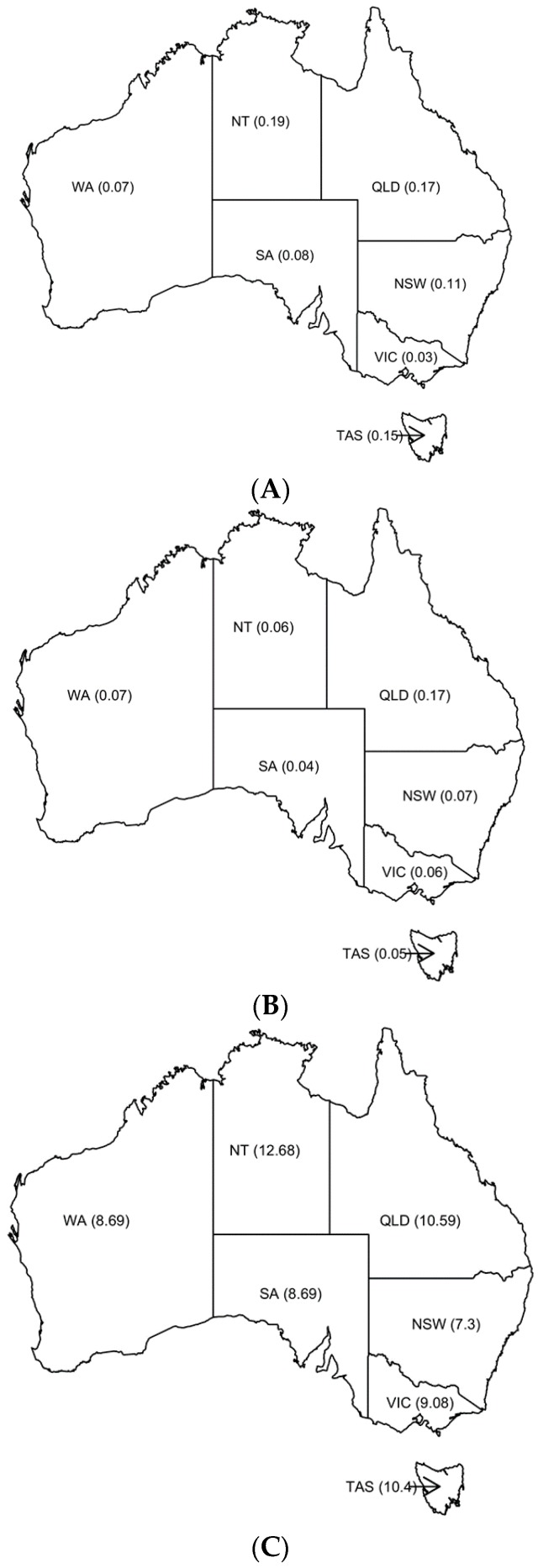
Numbers per capita (1000) of relinquished dogs on Gumtree (**A**) and PetRescue (**B**) from 8 to 24 February 2016, and total dog surrenders from Chua, Rand and Morton 2017 [2] (**C**) for each Australian State/Territory. Abbreviations used are: New South Wales (NSW), Victoria (Vic), Queensland (Qld), Western Australia (WA), South Australia (SA), Tasmania (Tas), Australian Capital Territory (ACT) and Northern Territory (NT)

**Figure 3 animals-08-00025-f003:**
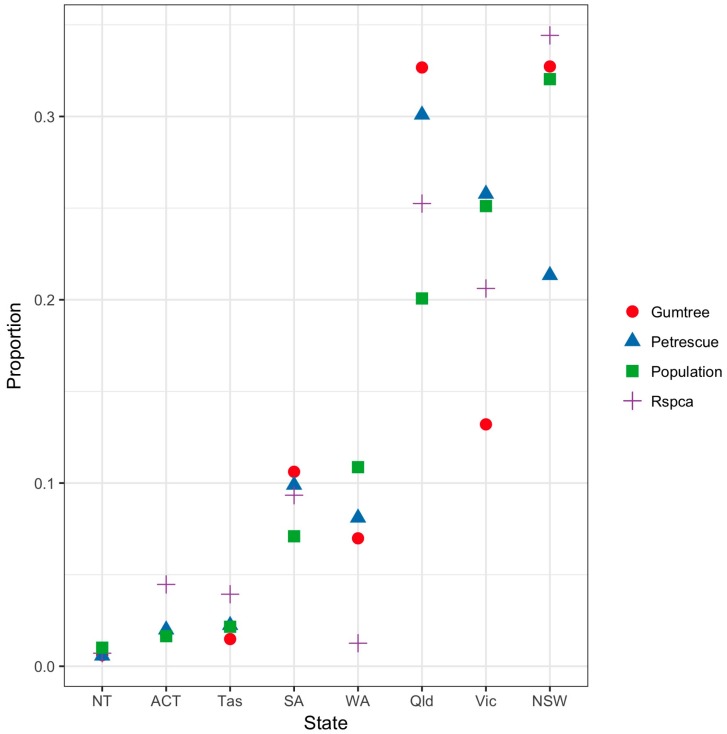
Proportions of total ads for relinquished cats on Gumtree and PetRescue from 9 to 26 February 2016, and Royal Society for the Prevention of Cruelty to Animals (RSPCA) shelter admissions for each Australian State/Territory. States/Territories are ordered in increasing proportion of total population. The square symbol is the proportion of the Australian population for each State/Territory (Note: The proportion of the Australian population in each State/Territory is: NSW 31.7%, Vic 24.9%, Qld 19.9%, WA 10.8%, SA 7%, Tas 2.1%, ACT 1.6% and NT 1%).

**Figure 4 animals-08-00025-f004:**
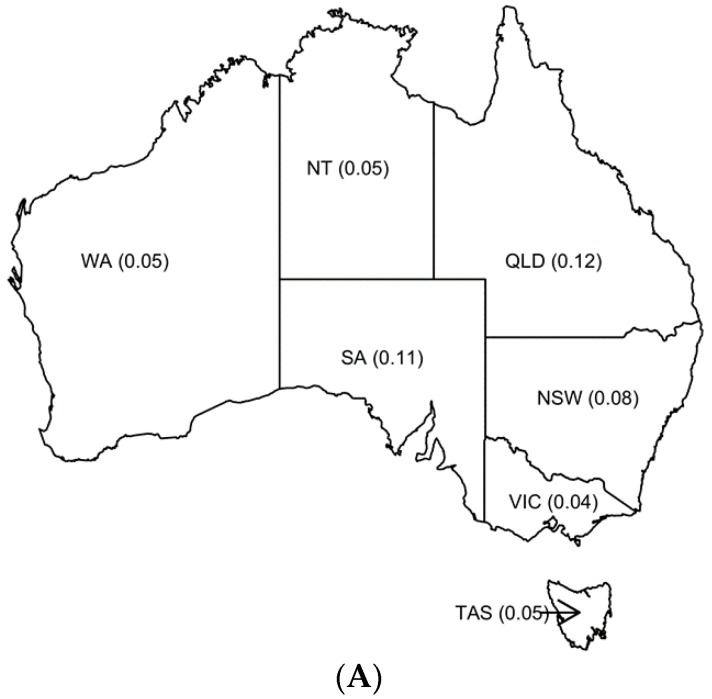
Numbers per capita (1000) of total ads for relinquished cats on Gumtree (**A**) and PetRescue (**B**) from 9 to 26 February 2016, and RSPCA shelter admissions (**C**) for each Australian State/Territory.

**Table 1 animals-08-00025-t001:** Dog breeds in which 20 or more individuals were relinquished on Gumtree or Pet Rescue from 8 February to 24 February 2016 (In 103 dogs the purebred status was unknown).

		Gumtree				PetRescue		
	# ^1^	Total	Pure	Cross	# ^1^	Total	Pure	Cross
Staffordshire Terrier	1	325 (12.3)	163 (6.1)	162 (6.1)	1	234 (11.5)	48 (9.9)	186 (12.1)
Kelpie	2	219 (8.3)	119 (4.5)	100 (3.8)	2	181 (8.9)	40 (8.3)	141 (9.1)
American Staffordshire Terrier	3	182 (6.9)	132 (5)	50 (1.9)	3	91 (4.5)	14 (2.9)	77 (5)
Bull Arab	4	135 (5.1)	37 (1.4)	98 (3.7)	7	74 (3.7)	7 (1.4)	67 (4.3)
Border Collie	5	120 (4.5)	64 (2.4)	56 (2.1)	5	85 (4.2)	11 (2.3)	74 (4.8)
Mastiff	6	103 (3.9)	28 (1.1)	75 (2.8)	9	64 (3.2)	4 (0.8)	60 (3.9)
Rottweiler	7	101 (3.8)	73 (2.8)	28 (1)	15	37 (1.8)	7 (1.4)	30 (1.9)
German Shepherd	8	98 (3.7)	74 (2.8)	24 (0.9)	12	50 (2.5)	24 (5)	26 (1.7)
Siberian Husky	9	87 (3.3)	66 (2.5)	21 (0.8)	19	25 (1.2)	18 (3.7)	7 (0.5)
Great Dane	10	77 (2.9)	26 (1)	51 (1.9)	13	50 (2.5)	10 (2.1)	40 (2.6)
Labrador	11	75 (2.8)	34 (1.3)	41 (1.6)	10	64 (3.2)	17 (3.5)	47 (3.6)
Cattledog	12	71 (2.7)	45 (1.7)	26 (1)	6	84 (4.1)	9 (1.9)	75 (4.9)
Bulldog	13	62 (2.3)	52 (2)	10 (0.4)	N	0	0	0
Chihuahua	14	55 (2.1)	39 (1.5)	16 (0.6)	16	30 (1.5)	8 (1.7)	22 (1.4)
Jack Russell Terrier	15	53 (2)	26 (1)	27 (1)	8	74 (3.7)	16 (3.3)	58 (3.8)
American Bulldog	16	42 (1.6)	27 (1)	15 (0.6)	21	20 (1)	1 (0.2)	19 (1.2)
Maltese	17	41 (1.6)	12 (0.5)	29 (1.1)	11	63 (3.1)	9 (1.9)	54 (3.5)
Bull Terrier	18	38 (1.4)	19 (0.7)	19 (0.7)	25	13 (0.6)	6 (1.2)	7 (0.5)
Shar Pei	19	35 (1.3)	18 (0.7)	17 (0.6)	22	20 (1)	3 (0.6)	17 (1.1)
Fox Terrier	20	35 (1.3)	21 (0.8)	14 (0.5)	18	28 (1.4)	5 (1)	23 (1.5)
Pomeranian	21	31 (1.2)	23 (0.9)	8 (0.3)	23	18 (0.9)	4 (0.8)	14 (0.9)
Malamute	22	27 (1)	21 (0.8)	6 (0.2)	26	13 (0.6)	6 (1.2)	7 (0.5)
Blue Heeler	23	25 (0.9)	12 (0.5)	13 (0.5)	27	5 (0.2)	1 (0.2)	4 (0.3)
Greyhound	24	25 (0.9)	23 (0.9)	2 (0.1)	3	109 (5.4)	98 (20.2)	11 (0.7)
Rhodesian Ridgeback	25	24 (0.9)	5 (0.2)	19 (0.7)	17	30 (1.5)	0	30 (1.9)
Maremma	26	20 (0.7)	17 (0.6)	3 (0.1)	28	4 (0.2)	1 (0.2)	3 (0.2)
Irish Wolfhound	27	20 (0.7)	2 (0.1)	18 (0.7)	24	14 (0.7)	1 (0.2)	13 (0.8)
Poodle	28	19 (0.7)	15 (0.6)	4 (0.2)	20	23 (1.1)	10 (2.1)	13 (0.8)
Total		2640 (100)	1357 (51.4)	1180 (44.7)		2027 (100)	484 (23.8)	1543 (75.8)

^1^ Ranking of breeds from most common to least common; N: not relevant.

**Table 2 animals-08-00025-t002:** Age and price categories of dogs relinquished on Gumtree from 8 to 24 February 2016.

Age Range	N (%)	Price Range (AUS $)	N (%)
Up to 16 weeks	20 (0.8)	0	575 (23.1)
16 weeks to 6 months	131 (5)	1–499	1181 (47.4)
6 months to 1 year	680 (26.1)	500–999	480 (19.3)
1 to 2 years	832 (32)	1000–1999	187 (7.5)
2 to 5 years	679 (26.1)	2000–2999	44 (1.8)
>5 years	260 (10)	3000–7000	26 (1)
Total	2602 (100)	Total	2493 (100)

**Table 3 animals-08-00025-t003:** Price range (AUS$) of dogs advertised on Gumtree between 7 and 26 February 2016 and their Australian State or Territory. Percentages represent the proportion of dogs within that State/Territory in each price category.

	$0 N (%)	$1–499 N (%)	>$500 N (%)
NSW	189 (23.2)	420 (50.2)	217 (26.6)
Qld	276 (34.5)	331 (41.4)	192 (24)
Vic	51 (24.8)	57 (27.7)	98 (47.6)
WA	43 (22.3)	79 (40.9)	71 (36.8)
SA	39 (28.5)	64 (46.7)	34 (24.8)
Tas	24 (30)	43 (53.8)	13 (16.3)
NT	15 (31.9)	22 (46.8)	10 (21.3)
Total	637	1006	635

**Table 4 animals-08-00025-t004:** Cat breeds relinquished on Gumtree from 9 to 26 February 2016 in comparison to those advertised in the same period on PetRescue.

		Gumtree				PetRescue		
	# ^1^	Total	Pure	Cross	#	Total	Pure	Cross
Ragdoll	1	148	116	32	4	30	24	6
DSH	2	81	0	81	1	1894	0	1894
Bengal	3	37	28	9	12	2	0	2
Persian	4	32	12	20	8	8	5	3
Siamese	5	29	19	10	5	16	15	1
Russian Blue	6	26	14	12	6	12	12	0
Burmese	7	24	19	5	8	8	8	0
British Shorthair	8	16	15	1	9	6	5	1
Manx	9	14	11	3	7	9	9	0
Birman	10	9	9	0	10	5	5	0
Himalayan	11	9	7	2	N	0	0	0
Devon Rex	12	9	7	2	N	0	0	0
Oriental	13	8	5	3	7	9	4	5
Tonkinese	14	8	6	2	11	3	2	1
Domestic Medium Hair	15	7	0	7	2	375	0	375
Maine Coon	16	5	5	0	12	2	2	0
Domestic Long Hair	17	4	0	4	3	38	0	38
Chinchilla	18	4	4	0	13	1	1	0
Sphynx	19	4	3	1	N	0	0	0
Snowshoe	20	0	0	0	11	3	2	1
Australian Mist	21	0	0	0	10	5	1	4
Total		508	310	198		2426	95	2331

^1^ Ranking of breeds from most common to least common; N = no rank as not present.

**Table 5 animals-08-00025-t005:** Price and age ranges of kittens and cats sold on Gumtree from 9 to 26 February 2016. There were 11 ads in which no age was provided and 90 ads in which no price was provided.

Age Range	N (%)	Price Range (AUS $)	N (%)
Up to 16 weeks	492 (23.6)	0	1298 (62)
16 weeks to 6 months	414 (19.9)	1–499	617 (29.5)
6 months to 1 year	201 (9.7)	500–999	73 (3.5)
1 to 2 years	373 (17.9)	1000–1999	13 (0.6)
2 to 5 years	391 (18.8)	2000–2999	2 (0.1)
>5 years	211 (10.1)		
Total	2082 (100)	Total	2003 (100)

**Table 6 animals-08-00025-t006:** Price range of cats advertised on Gumtree between 7 and 26 February 2016 and their Australian State or Territory. Percentages represent the proportion of cats within that State/Territory in each price category.

	$0 N (%)	$1–499 N (%)	>$500 N (%)
New South Wales	356 (62.2)	190 (33.2)	26 (4.5)
Queensland	366 (64.3)	183 (32.2)	20 (3.5)
Victoria	178 (77.1)	36 (15.6)	17 (7.4)
Western Australia	65 (53.7)	50 (41.3)	6 (5)
South Australia	125 (67.2)	56 (30.1)	5 (2.7)
Total	1090 (64.9)	515 (30.7)	74 (4.4)

**Table 7 animals-08-00025-t007:** Demographic factors for the dogs and cats being relinquished by the 16 people participating in surveys of why they relinquished their pets on Gumtree. Responses to the pet’s Health and Behavior were scored on a 1–5 Likert-type scale where 1 was very dissatisfied and 5 was very satisfied.

Age (yrs)	Sex	Breed	Neut. ^1^	MC^2^	Health	Behav. ^3^	Original Source
**Dogs**							
1	M	Pug x Jack Russell Terrier	NS	NS	NS	NS	Gumtree (rescue)
4	F	Greyhound	No	Yes	NS	NS	NS
1	F	Bull Mastiff X	No	Yes	5	NS	Gumtree
7	F	Siberian Husky	Yes	Yes	4	3–3.5	Friend as puppy
6	M/F	Maltese/Maltese X Pomeranian	Yes	Yes	4 & 5	5 & 5	TradingPost
NS	M/F	Beagle	Yes	NS	5 & 5	5 & 5	Gumtree (rescue)
4.5	M	Jack Russell	Yes	Yes	5	3	Registered breeder
1.5	M	American Staffordshire Terrier	Yes	Yes	5	5	Unregistered breeder
**Cats**							
1	M	Ragdoll X	Yes	Yes	5	5	Gumtree
0.33	F	NS	No	Yes	NS		Gumtree
4.5	M	DSH	Yes	Yes	5	2	Family litter
3	M	British Shorthair	Yes	Yes	5	5	NS
2	M	DSH	Yes	No	5	NS	Family with 2 Ragdolls
0.92	M	Ragdoll	Yes	Yes	5	3	Family
NS	NS	DSH	NS	NS	NS	NS	Strays

^1^ Neutered; ^2^ Microchipped; ^3^ Behavior; NS = not stated; x = cross-breed; M = male; F = female.

**Table 8 animals-08-00025-t008:** Reasons for relinquishment given by dog and cat owners advertising their pet on Gumtree.

Dogs	N	Cats	N
Rescue	2	Dog/cat not getting on	3
Behaviour	2	Moving	2
Dog injury (working dog)	1	Rescue	1
Own health	1	Allergy	1
Money	1	Expecting a baby	1
Relationship breakdown	1	Too many pets	1
Moving	1	Time	1
Time	1		

**Table 9 animals-08-00025-t009:** Reasons people gave for choosing to advertise their relinquished pet on Gumtree.

Dogs	N	Cats	N
Shelters full	4	Got the animal on gumtree	2
To see/interview the new owner	2	Use gumtree for other things	2
Lack of knowledge of shelters	1	To see and interview the new owner	2
It worked before	1	It worked before	1
Shelter is last resort	1	Shelter is last resort	1
Immediate posting and quick response	1	Would be put down at a shelter	1
Would be put down at a shelter	1	Shelter too hard on pet	1
		Don't drive so can’t get to shelter	1

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
