# Peer review of "Online Relinquishments of Dogs and Cats in Australia"

_animals, 2018, doi:10.3390/ani8020025_

Round 1

Reviewer 1 Report

The authors address an important gap in the understanding of the flow of dogs and cats through and around owners, breeders, shelters, and the street. Monitoring has been largely put in place in shelters, but the flow of animals that bypass shelters via the Internet is largely unknown. This manuscript describes one of the first studies to fill this gap.

The work is well-done, good design, sample size, and analysis. Interpretation is appropriate and documented well. The writing is excellent and I did not identify any typographical errors.

My one disappointment is in the small sample size for the follow-up interviews. But the rate of reply is typical, and I can imagine that many people would not be interested in participating for emotional reasons.

I recommend this manuscript for publication in its current form.

Author Response

We thank Reviewer 1 for these comments.

Reviewer 2 Report

This paper explores online trading of companion animals in Australia via html-scraping methodologies, and contributes to the limited research into this area. 

Unfortunately, one of the reasons why research into this field is limited is that there are considerable ethical concerns with html-scraping or analysing publicly available data. The maxim "Just because it's public does not mean its available" should be borne in mind. I would advise the authors seek out and study the Association for Internet Researchers (AOIR)'s ethical guidelines (https://aoir.org/ethics/) and they must explore the ethical implications of their work, particularly in light of the very low response rate and the fact several participants contacted them to say the work was distressing. 

The editor must decide whether they feel it is appropriate to publish this work. My personal ethical position on this type of research is that it is extremely valuable, but we must take into consideration the fact that the people in your study did not consent to have their information mined for data like this, were given no way to opt out of the study, and may indeed have motivations which would have disincentivised them from participating (e.g. were operating an illegal or immoral breeders). And these considerations must be discussed in the manuscript.

I also have a personal bugbear about the methodology. I believe that both the scraping and interviews have not truly made the best use of methodologies available. I will not necessarily demand that these be addressed, but I believe the paper would be much stronger and more interesting if at least one of these alternatives is considered: Given the html-scraping, one way of finding information out while still preserving participants' anonymity would be to explore text mining (e.g. how are words associated in the dataset). There are a number of R packages that can help with this, e.g. 'tm', although I personally advocate the use of 'textstem' over 'tm's inbuilt stemming functions should you wish to do that. This can provide a richer exploration of patterns without identifying people, helping to eliminate some of the AOIR ethical concerns. Within the survey data, I do not believe that the kind of information obtained from the 15 respondents is very meaningful, and it would be greatly improved if you could explore motivations in more detail in a qualitative analysis. If any of your participants would be willing to be contacted again for more detailed exploration of why they used Gumtree, and their thoughts about the people who contacted them, you would have more interesting findings. 

Finally - while reading the methodology I had many questions which were later answered in the results. It's very tricky with this kind of 'novel data source' study to decide what information should be presented where, but I certainly felt frustrated at having to wait so long for some pieces of information. Consider revising (I'll note where I was asking questions). 

Line by line comments:

Simple Summary: Although the simple summary does not need to have statistics, it should clearly indicate what information came from this study and what came from elsewhere. 

Line 11: Estimated by this study?

Line 12: how many more cats than dogs? (In addition, the sentence construction implies that median age affected price. Split this sentence). 

Line 16 - How did you come to this result (via interview we explored reasons for relinquishment?)

Line 22 - Data was collected in 2016. Dogs were sampled on February 7th, 16th, etc..

Line 22: Here I wanted to know how many ads provided the age of their dog and cat

Line 26: Free as in the cost of the advert or free as in the animal was free?

Line 40: Why is this in addition to the previous point? I see no connection between them.

Lines 88-92: Much of the intro and discussion is repetitive, this section particularly. With the new ethical discussions you will be bringing in you'll be able to cut some of this back 

Line 116: Give version number (or release data of version you used) for BeautifulSoup.

Here I wanted to know how many ads were scraped. 

Line 125: Here I wanted to know many ads and what proportion of ads were sampled to generate the number per ad? Were many ads advertising more than one dog/cat? How many?

Line 137: Were any ads from 'rehomers', e.g. a small shelter that didn't want to bother with their own website?

Line 153: Here I wanted to know the response rate

Line 177: Given that many criticisms of selling animals on gumtree come from the worry that there are disreputable breeders posting repeated 'oh an accidental litter' ads, I think this extrapolation may not be very reliable. It may be an underestimate or an over estimate, and I would be nice to get some discussion of this, and to draw on confidence intervals reported elsewhere (these don't have to be from animal studies)

Line 304: I am really concerned that some of your would-be participants were too distressed to respond. You must report how many were too distressed and what support you offered them (and if it was none, you must say it was none and then reflect on this for future studies)

Results General: With a response rate this low I doubt the relevance of the reported survey data and would prefer a more qualitative exploration of the participants thoughts and feeling

Line 481-486: Is this not just a socially acceptable thing to say? (See Izuma 2012 http://dx.doi.org/10.1016/j.neures.2012.01.003 and the discussion about how all online behaviour is 'observed'). 

Overall, and especially in light of the ethical concerns, the discussion is sorely missing an exploration of human motivation and online behaviour

Author Response

This paper explores online trading of companion animals in Australia via html-scraping methodologies, and contributes to the limited research into this area.

Unfortunately, one of the reasons why research into this field is limited is that there are considerable ethical concerns with html-scraping or analysing publicly available data. The maxim "Just because it's public does not mean its available" should be borne in mind. I would advise the authors seek out and study the Association for Internet Researchers (AOIR)'s ethical guidelines (https://aoir.org/ethics/) and they must explore the ethical implications of their work, particularly in light of the very low response rate and the fact several participants contacted them to say the work was distressing.

We thank the reviewer for these comments, as the ethical questions that arise in performing internet research are important to address. We have reviewed the AOIR ethical guidelines and do not believe that any of the methods used in this research contravene the ethical principles outlined in these guidelines. We obtained ethical approval from the Human Research Ethics Committee of the University of Adelaide, and the only contact made with potential participants was via an email that outlined the purpose of the study. Only people who consented to be part of the research were contacted by the researchers, either by email or phone depending on the choice of the participant. The people who emailed to state that they did not wish to be involved were not contacted further as this would have contravened the ethical approval which was given.

The editor must decide whether they feel it is appropriate to publish this work. My personal ethical position on this type of research is that it is extremely valuable, but we must take into consideration the fact that the people in your study did not consent to have their information mined for data like this, were given no way to opt out of the study, and may indeed have motivations which would have disincentivised them from participating (e.g. were operating an illegal or immoral breeders). And these considerations must be discussed in the manuscript.

A paragraph has been added to the discussion which includes the ethical considerations of the research.

With respect to the data obtained from web scraping, all of this is kept on password protected computers belonging to employees of the University of Adelaide. Only composite data (e.g. numbers of dogs and cats, breed and price) for the ads are included in the research paper, and the dataset is not publicly available. The quotes used in the paper are from the website and publicly available at the time the ads were current. Since the ads were up over a year ago it is highly unlikely any could be traced back to the owner of the animal. There are no significant harms or risks for the owners whose ads have been scraped as they are not identifiable from the material published. We believe the potential benefits of the project in highlighting the large number of dogs and cats being relinquished online outweigh any perceived risks relating to the research methods used.

I also have a personal bugbear about the methodology. I believe that both the scraping and interviews have not truly made the best use of methodologies available. I will not necessarily demand that these be addressed, but I believe the paper would be much stronger and more interesting if at least one of these alternatives is considered: Given the html-scraping, one way of finding information out while still preserving participants' anonymity would be to explore text mining (e.g. how are words associated in the dataset). There are a number of R packages that can help with this, e.g. 'tm', although I personally advocate the use of 'textstem' over 'tm's inbuilt stemming functions should you wish to do that. This can provide a richer exploration of patterns without identifying people, helping to eliminate some of the AOIR ethical concerns. Within the survey data, I do not believe that the kind of information obtained from the 15 respondents is very meaningful, and it would be greatly improved if you could explore motivations in more detail in a qualitative analysis. If any of your participants would be willing to be contacted again for more detailed exploration of why they used Gumtree, and their thoughts about the people who contacted them, you would have more interesting findings.

Again we thank the reviewer for these suggestions. We have considered the use of text mining using R, but this is outside the scope of the current study that aimed to find out the numbers and demographics of dogs and cats relinquished online. We may explore text mining in further studies. With respect to the participants that were interviewed, there was no question included on whether they agreed to be contacted again for further questions and so this would not be possible with the ethics approval that was given. We agree that the information from the 15 respondents is likely to be biased, as we discuss in the paper. However, we think the information provided gives some valuable information that can be used as a basis for future research.

Finally - while reading the methodology I had many questions which were later answered in the results. It's very tricky with this kind of 'novel data source' study to decide what information should be presented where, but I certainly felt frustrated at having to wait so long for some pieces of information. Consider revising (I'll note where I was asking questions).

Revisions have been made in response to the following comments.

Line by line comments:

Simple Summary: Although the simple summary does not need to have statistics, it should clearly indicate what information came from this study and what came from elsewhere.

All of the information in the Simple Summary comes from the study itself. We have removed the extrapolation to the numbers of dogs and cats relinquished per year and replaced it with the numbers of ads identified during February 2016.

Response

Line 11: Estimated by this study?

We have removed the extrapolation and replaced it with the numbers of ads identified during February 2016.

Line 12: how many more cats than dogs? (In addition, the sentence construction implies that median age affected price. Split this sentence).

The wording has been changed to ‘There were 23% of dog ads and 62% of cat ads for free animals. The median age was 1.42 years in dogs and 0.9 years in cats.’

Line 16 - How did you come to this result (via interview we explored reasons for relinquishment?)

Wording has been added: ‘Fifteen people who had relinquished a dog or cat on gumtree.com.au were interviewed.’

Line 22 - Data was collected in 2016. Dogs were sampled on February 7th, 16th, etc..

The wording has been changed as suggested.

Line 22: Here I wanted to know how many ads provided the age of their dog and cat

All ads on gumtree.com.au provide a date of birth for the pet.

Line 26: Free as in the cost of the advert or free as in the animal was free?

Thanks for picking this up, the wording has been clarified.

Line 40: Why is this in addition to the previous point? I see no connection between them.

The ‘In addition..’ links to the previous sentence as the flow between populations of dogs and cats is even more complicated than simply between owned and unowned populations, and also includes  different owners.

Lines 88-92: Much of the intro and discussion is repetitive, this section particularly. With the new ethical discussions you will be bringing in you'll be able to cut some of this back

The first sentences in this paragraph have now been removed.

Line 116: Give version number (or release data of version you used) for BeautifulSoup.

The version and release date have been added.

Here I wanted to know how many ads were scraped.

All of the ads in the dog and cat sections were scraped on the dates stated. Many of the ads were unrelated to the research questions (e.g. for pet sitting, or pet collars or harnesses) and these were removed early in the cleaning of the data.

Line 125: Here I wanted to know many ads and what proportion of ads were sampled to generate the number per ad? Were many ads advertising more than one dog/cat? How many?

We have added ‘There were 50 random ads used for both dogs (1.9% of total ads) and cats (2.4% of total ads).’ The information on the number of ads advertising more than one dog/cat has been added to the Results section.

Line 137: Were any ads from 'rehomers', e.g. a small shelter that didn't want to bother with their own website?

In this section we have added: ‘On gumtree.com.au sellers are classified as owner, breeder or shelter and the shelter animals were not included in the present study. It is possible that some very small shelters were advertising as an owner rather than as a shelter, but there was no way to differentiate these types of sellers. Both owner and breeder categories were included as there were some breeders who were relinquishing pets using the above criteria…’

Line 153: Here I wanted to know the response rate

This is already included in the Results section on Line 302: ‘There were 15 responses in total, a response rate of 2.5% (8/324) for dogs and 2.3% (7/299) for cats.’

Line 177: Given that many criticisms of selling animals on gumtree come from the worry that there are disreputable breeders posting repeated 'oh an accidental litter' ads, I think this extrapolation may not be very reliable. It may be an underestimate or an over estimate, and I would be nice to get some discussion of this, and to draw on confidence intervals reported elsewhere (these don't have to be from animal studies)

Line 177 gives the total number of ads for dogs on gumtree and I’m not sure how that relates to what is written above. I’m guessing it’s the extrapolation from the numbers of ads posted on the dates in February to the total numbers of ads that would be advertised in a year. It is acknowledged that this extrapolation may not be accurate as the numbers of ads in different months may vary. We have added this as a limitation in the final paragraph before the Conclusion. We don’t see how disreputable breeders posting ads is likely to impact the extrapolation made- if the reviewer can clarify what they mean we will address their concern.

Line 304: I am really concerned that some of your would-be participants were too distressed to respond. You must report how many were too distressed and what support you offered them (and if it was none, you must say it was none and then reflect on this for future studies)

As part of the ethical approval we could only contact people who had consented to participate in the research. The people who responded that they were too upset to participate could not be contacted for this reason. Thus we could not email offering any support. We had not expected to get responses from people who did not want to participate in the research- this has not happened in our previous research.

Results General: With a response rate this low I doubt the relevance of the reported survey data and would prefer a more qualitative exploration of the participants thoughts and feeling

The type of exploration of the respondents who had relinquished pets is more qualitative than quantitative in nature, with numbers only given to illustrate the themes that emerged from their responses. Demographic data is provided to give background on the respondents.

Line 481-486: Is this not just a socially acceptable thing to say? (See Izuma 2012 http://dx.doi.org/10.1016/j.neures.2012.01.003 and the discussion about how all online behaviour is 'observed').

We have added the need for social acceptability and possibility of providing false reasons for the relinquishment in the final limitations paragraph in the Discussion.

Overall, and especially in light of the ethical concerns, the discussion is sorely missing an exploration of human motivation and online behaviour

A paragraph on the ethical concerns has been added to the Discussion. However, we have failed to find scientific studies on the motivation of people to obtain pets online, or on their online behaviour. If the reviewer can suggest some studies to cite we will gladly add this to the Discussion. 

Reviewer 3 Report

44: Animals leave the care of their owners through lost (accidental stray)

55: Relinquished typically indicates animals that are owner-surrendered; recommend changing "relinquished" to "admitted" for consistency with source data and US terminology.  Consider referencing the other major categories of outcome in referenced source, including those rehomed and euthanized.  Consider discussing proportion of reclaimed animals versus rehomed or euthanized (in the US, it would be uncommon for the reclaim rate to be so high as compared to other outcomes).

169: Table does not seem useful

202: While the result is statistically significant, it does not seem to be clinically meaningful.  If it is clinically meaningful, please provide further discussion.

209:  Data may be better displayed as a graph with continuous data rather than a table with binned data

211-224: data may be better expressed as a per capita rate (# per 1,000 residents).  Consider geographically weighted regression, at least as a future investigation in the discussion section.  This may be particularly useful to reader unfamiliar with Australian geography.

225: Data may be better expressed as a series of choropleth maps (one per online marketplace) based on per capita rather than proportion.  In black and white, unable to discriminate between online marketplaces (all look like filled circle).

236: Consider expressing these data as a map with mean or median value.

262: While the result is statistically significant, it does not seem to be clinically meaningful.  If it is clinically meaningful, please provide further discussion.

285: Data may be better expressed as a series of choropleth maps (one per online marketplace) based on per capita rather than proportion.  In black and white, unable to discriminate between online marketplaces (all look like filled circle).

356: Expand on observed differences, consider discussing theories as to why, which would be particularly useful to those not familiar with these marketplaces and geography.

358-370: Discuss limitation of time of year when extrapolating from such a small window of time, particularly for cats which are seasonal breeders.

381: Consider further discussion about why Queensland may be special (ideally, geographically weighted regression might point to some predictors).

439: In a more recent article, this may not be true for cats.  Weiss, Emily, and Shannon Gramann. "A comparison of attachment levels of adopters of cats: Fee-based adoptions versus free adoptions." Journal of Applied Animal Welfare Science 12.4 (2009): 360-370.

Author Response

44: Animals leave the care of their owners through lost (accidental stray)

Lost animals has been added to this sentence.

55: Relinquished typically indicates animals that are owner-surrendered; recommend changing "relinquished" to "admitted" for consistency with source data and US terminology.  Consider referencing the other major categories of outcome in referenced source, including those rehomed and euthanized.  Consider discussing proportion of reclaimed animals versus rehomed or euthanized (in the US, it would be uncommon for the reclaim rate to be so high as compared to other outcomes).

The word ‘relinquished’ has been changed to ‘admitted’ in this sentence. The other additions have not been added as although interesting information they are outside the focus of the present study. 

169: Table does not seem useful

The table has been removed at the suggestion of the Reviewer.

202: While the result is statistically significant, it does not seem to be clinically meaningful.  If it is clinically meaningful, please provide further discussion.

The results may not be clinically significant, but are included for interest for the reader. Since the data were not normally distributed and a non-parametric test was used, we have also included the median values.

209:  Data may be better displayed as a graph with continuous data rather than a table with binned data

We believe the presentation of the data as categories makes it easier for the reader to work out the percentage of dogs/puppies in each category. The high proportion of ads for free dogs/puppies would also make a figure less appealing to view.

211-224: data may be better expressed as a per capita rate (# per 1,000 residents).  Consider geographically weighted regression, at least as a future investigation in the discussion section.  This may be particularly useful to reader unfamiliar with Australian geography.

We have included a second Figure as suggested by the reviewer which has per capita figures and chloropleth maps (see response below), but Figure 1 is given as a proportion as this better indicates the patterns across Australia. The population size is given for each state to indicate the state sizes for those not familiar with Australia. To account for spatial dependence, in future research we will consider a geographically weighted regression. At present, we are merely considering a hypothesis generating approach.

225: Data may be better expressed as a series of choropleth maps (one per online marketplace) based on per capita rather than proportion.  In black and white, unable to discriminate between online marketplaces (all look like filled circle).

We thank the reviewer for this suggestion, and have added a chloropleth maps as suggested for each marketplace. These demonstrate some additional patterns that are not seen when using the proportions between States/Territories. We believe that many people now look at papers online, and that the colour used in the existing figure is adequate to see the different marketplaces.

236: Consider expressing these data as a map with mean or median value.

We believe that expressing these data as a map with values per price category per State/Territory would make it more difficult for the reader to evaluate the data and that the Table is preferable.

262: While the result is statistically significant, it does not seem to be clinically meaningful.  If it is clinically meaningful, please provide further discussion.

The results may not be clinically significant, but are included for interest for the reader. Since the data were not normally distributed and a non-parametric test was used, we have also included the median values.

285: Data may be better expressed as a series of choropleth maps (one per online marketplace) based on per capita rather than proportion.  In black and white, unable to discriminate between online marketplaces (all look like filled circle).

See the responses for the dog data above- the same changes have been made for the cat data.

356: Expand on observed differences, consider discussing theories as to why, which would be particularly useful to those not familiar with these marketplaces and geography.

We do have a paragraph in which a comparison between the different populations (Gumtree, RSPCA and PetRescue) is made. There are many variables we do not know, such as the true number of dog and cat owners in each state/territory, and the proportion who relinquish a pet in each state/territory. We feel that to expand the Discussion as to other reasons for the differences would involve more conjecture than argument backed up by validated observations and have refrained from this. If the reviewer can provide any other studies with evidence that could be used we would be happy to include them.

358-370: Discuss limitation of time of year when extrapolating from such a small window of time, particularly for cats which are seasonal breeders.

This has been added to a final paragraph on limitations of the study, the final paragraph before the Conclusion.

381: Consider further discussion about why Queensland may be special (ideally, geographically weighted regression might point to some predictors).

As stated above we could not find any other validated studies to explain the difference between states. Although I can think of reasons they would be pure conjecture, and I think the results should be used in further studies to try to determine the reasons.

439: In a more recent article, this may not be true for cats.  Weiss, Emily, and Shannon Gramann. "A comparison of attachment levels of adopters of cats: Fee-based adoptions versus free adoptions." Journal of Applied Animal Welfare Science 12.4 (2009): 360-370.

We have added a sentence acknowledging the differing conclusion from this study. 

Round 2

Reviewer 2 Report

Overall I think this paper is much clearer after revision, and the revised methods were less frustrating to read  with the info supplied more immediately. 

I want to highlight to the authors that I think this is valuable work. I think your ethical discussion is very suitable

Figure 3 - what is the significance of squares vs circles? I think it's that squares are the population but this isn't clear in the legend

Author Response

We thank for reviewer for their comments.

In both Figure 1 and Figure 3 the square symbol is the proportion of the State/Territory population. This has been added to the figure legends.

Reviewer 3 Report

25: "data" is generally considered to be the plural form, so I believe that "were" may be more appropriate here than "was".  However, I leave the final determination (here and where subsequently found) to the editorial process.

122: Should be "ads" rather than "adverts" per the abbreviation established on line 24.

123: Give Excel version

251 and subsequent choropleth maps: Thank you for adding the choropleth maps, I found them to be helpful.  However, with continuous color schemes the lighter shades are more typically used for lower data values, and darker for higher http://pro.arcgis.com/en/pro-app/help/mapping/layer-properties/graduated-colors.htm . 

Author Response

We thank the reviewer for their comments.

Our responses follow:

25: "data" is generally considered to be the plural form, so I believe that "were" may be more appropriate here than "was".  However, I leave the final determination (here and where subsequently found) to the editorial process.

We will leave it to the editors to make any changes necessary.

122: Should be "ads" rather than "adverts" per the abbreviation established on line 24.

This has been changed.

123: Give Excel version

The version has been added.

251 and subsequent choropleth maps: Thank you for adding the choropleth maps, I found them to be helpful.  However, with continuous color schemes the lighter shades are more typically used for lower data values, and darker for higher http://pro.arcgis.com/en/pro-app/help/mapping/layer-properties/graduated-colors.htm .

We have reversed the colour coding on all of the maps so that the darker colours represent a higher density. Thanks this does make the maps easier to interpret.